# Support measures for the next of kin who has experienced the unexpected loss of a family member to HIV/AIDS

**Siphesihle Delani Hlophe** [ID]◐*, **Karien Jooste**◐

Department of Nursing Science, Faculty of Health and Wellness Sciences, Cape Peninsula University of Technology, Cape Town, South Africa

◐ These authors contributed equally to this work.
* nursesiphe@gmail.com

## Abstract

**Data Availability Statement:** Data can There are no restrictions to accessing study data; the URL to the repository is as below and is accessible by the public: https://etd.cput.ac.za/handle/20.500.11838/ 3370 Research Ethics Committee: Ms

### Background

Passing of a close family member is one of the foremost traumatic occasions in a person's life. The way in which this misfortune unfolds, shifts from individual to individual, and depend on how close you were with the deceased. It was unclear on what were the support measures provided to youth after the loss a family member to HIV/AIDS.

### Aim

The aim of this article is to understand the support measures for the youth following the unexpected loss of a family member to HIV/AIDS.

### Setting

Khayelitsha, Western Cape province, South Africa.

### Method

A descriptive phenomenological design was followed, with an accessible population of youth who lost a family member to HIV/AIDS. Individual semi-structured interviews were conducted with 11 purposively selected participants after obtaining written informed consent. The sessions held with an interview schedule, did not take longer than 45 minutes to conduct until data saturation was reached. A digital recorder was used and field notes held. Open coding followed after transcribing interviews.

### Results

Youths did not know how to manage themselves as a result of a lack of therapeutic sessions, which could provide emotional support and help them with their healing process.

Nomathemba Seth, SethN@cput.ac.za Prof Karien
Jooste joosteka@cput.ac.za Tel: +27 21 959
6523be obtained by requesting it from the
corresponding author S.D.H., through the
Research Ethics Committee of the Cape Peninsula
University of Technology, or the open-access
thesis repository of the Cape Peninsula University
of Technology. No restrictions.

**Funding:** The authors received no specific funding
for this work.

**Competing interests:** The authors have declared
that no competing interests exist.

## Conclusion

Support measures for the next of kin were needed. Grieving influenced the emotions of an individual who experienced the burden of not having someone to speak to about their feelings.

## Contribution

The context-based information in this study addresses the important of support measures to be provided or implemented for the next of kin after they had lost a family member.

## Introduction

A family is often a source of emotional support, love, security, and protection and security, and may give an interesting sense of having a place and values that cannot be found in other relationships [1]. Youth going through the unexpected death of a family member, should be assisted with handling grief and setting up a positive intellect set around figuring it out their future. Youth are defined as persons between 18 and 25 years in the state or quality of being young, energetic, and immature [2].

According to Matthews [3], when it comes to grieving the death of a loved one, there are no linear patterns, no 'normal' reactions, and no formulas to follow. The reality of death affects families in a myriad of emotional/physical ways. Lekalakala-Mokgele [4] states that individual grief is unique and does not follow any pattern or path. Youth could have different ways of showing grief such as crying, fear, and anger which are common and universal; however, the cultural environment should also be considered when complicated grief is experienced. Most bereaved people can overcome their grief, but in some cases, grief becomes prolonged or complicated [5].

Individuals who lose a family member go through various stages. Stroebe, Schut and Boerner [6] stated that it can frequently be interpreted as a prescription of progress that bereaved persons must implement to get to terms with the loss. On the other hand, the authors mention that stages are often time-related, and each stage is necessary to reach the following stage [7]. Different individuals experience grief in different ways, not necessarily cycling through the stages to overcome their grief, and stages of grief can overlap. In this process of change, grieving people such as youth almost always need the support of others. The common thread for all grieving people is change [8]. Not only do they change inwardly, but their daily routines change leading to a measure of adjustment which can be hard to deal with.

Someone who unexpectedly loses a family member can move on a continuum range of emotions such as from denial to acceptance [9]. In the process of taking ownership of playing a new role in assisting others (who were previously supported by the deceased), they sometimes give up on their own dreams. Sad experiences could block out memories of good times they had with the deceased [10]. When you're grieving, it's more critical than ever *to take care of yourself*. The consequences of a traumatic incident can rapidly exhaust your energy and emotional reserves. Looking after one's physical and emotional needs will assist one to get through a troublesome time [11] and manage yourself.

Self-management, that *is the act of practising intentional self-care*, could thus help youth to move through different stages of grief, to get stronger, and fill themselves with hope for what lies ahead [12].

Lenzen, Daniels, Bokhoven, Van der Weijden and Beurskens [13] define self-management *as the degree to which persons have the ability to control their own lives and to cope effectively with adjustment, such as grief.* When a family member passes away successful self-management could be demonstrated by dealing with the stages of grief and towards improving one's health status [14]. The management of oneself could decrease distress associated with a life event such as losing a family member. Empirical evidence confirms that health outcomes are enhanced in individuals such as youth, who engage in self-management, using condition-specific knowledge, beliefs and self-regulation skills [14]. Youth are defined as persons between the ages of 18 and 25 years in the state or quality of being young, energetic and immature [15].

Going through stressful times and stages is part of life, however, youth may feel unfocused, overwhelmed and helpless [16]. Youth who have lost their parents may also increase their alcohol use and their relationship with siblings may be positively or negatively affected [17].

An active approach (including cognitive and behavioural strategies) develops self-management skills via self-reflection, solving problems and active goal setting [18]. Individuals must not discontinue practising self-management strategies, regardless of their emotional pain, as it is an ongoing healing process that is crucial for the development of the ability to separate the "self" from pain and to develop self-efficacy [19].

Practising intentional self-management can help family members to feel stronger, restore their sense of peace, and fill one with hope for what lies ahead [12]. In this study, it referred to the planning, organising, directing and control of the youth deal with after the loss of a family member, themselves.

It is essential to understand the experiences causing pain and grief to individuals to optimise the design and delivery of self-management interventions [19]. Youth could experience emotions such as fear and depression after the loss of a family member with e.g., HIV/AIDS. In some families, a young person must then act and behave as a parent when the actual parent passes away [17]. People who are experiencing grief are very likely to be depressed and consequently less likely to engage in active self-management strategies [19]. It was observed that some youth visiting a public comprehensive Primary Healthcare clinic in Cape Town, experienced panic attacks, anxiety and a loss of direction as a result of losing a family member. It was therefore unclear what they lived experiences were of youth, managing themselves after the loss of a family member to HIV/AIDS.

## Purpose

The purpose of this study was to explore the lived experiences of youth on self-management following the loss of a family member with HIV/AIDS.

## Methods

The design of the study was descriptive phenomenology and allowed the researcher to gain an in-depth understanding about the phenomenon of losing a family member to HIV/AIDS.

The accessible population was youth who visited a Comprehensive Primary Healthcare clinic in Cape Town, of the Western Cape in February 2020. Purposive sampling was followed, there was and inclusion criteria that the participants had to meet to be included in the study. The inclusion criteria were youths who had lost a close family member to HIV/AIDS, and who lived in the same household in a period of the previous six months. Eleven (n = 11) semi-structured individual interviews were conducted Interviews followed the inquiry method, using an interview guide, that lasted around 45 minutes. The utilisation of probing questions led to data saturation and yielded knowledge of the lived experiences of the youth.

## Data gathering

Participant recruitment took place at a Comprehensive Primary Healthcare clinic between January and February 2020. This was according to an agreement made with the professional nurse in charge. The invitation was in the form of a poster placed at the entrance of the Comprehensive Primary Healthcare clinic, inviting participants to participate in the study. After completing the health consultations, staff also referred patients (youth) to the waiting area outside a private room where the interviews were conducted. Participants were interviewed at a clinic room where there were no disturbances. The researcher asked participants if he could keep field notes and use a digital recorder during the interviews.

## Data analyses

The recordings on the digital recording were transcribed. Some of the interviews had to be translated. The translation was due to the use of a local language (isiXhosa) by a few participants; it was translated into English by the researcher and back-translated by an editor who also spoke both languages to ensure the dependability of the data interpretation. Data transcription began by establishing the unit of analysis to be studied, addressing which information should be included in a transcription. Interview data and field notes were coded simultaneously. An independent coder held a consensus meeting with the researcher on the themes and categories that emerged from the data.

## Trustworthiness

The trustworthiness of the study was ensured in various ways. Credibility was ensured through triangulation as it involved using different data collection methods of interviews and field notes to ensure the findings' consistency. Dependability was confirmed by using an independent coder who analysed the data and results of the study and had a consensus meeting with the researcher. To ensure confirmability, the data reflected the participants' voices, inquiry audit techniques, reflexivity (fairness in inclusion criteria), and triangulation of data (interviews and field notes). Transferability in the study was established by providing a thick description of the methodology and findings as evidence that the research study's results could be applied in the contexts, situations, times, and populations. The applicability of the results was limited to females.

## Ethical principles

The ethical clearance (permission) was obtained from the Research and Ethics Committee of the Faculty of Health and Wellness at Cape Peninsula University of Technology (Ethics clearance number: CPUT/HW-REC 2019/H2) and the Department of Health of the Western Cape Province, after applying on The National Health Research Database (Ethics clearance number: WC_201911_032). Participants received an information sheet in their language of choice (English, Xhosa). The study was verbally explained, and their questions were answered. Participants participated in the study voluntarily and could withdraw from the study at any time without any consequences or implications. Confidentiality was maintained as the names of participants in the interviews remained anonymous on the transcripts. Harm to the participant was minimal (some showed some emotions, crying); however, an advanced psychiatric nurse practitioner was arranged to be available at the clinic if a participant needed emotional support and had to be referred. Data were stored online with password-protected files on the main researchers' computers. It is planned that all data will be destroyed five years after the publication of the report of the study.

## Results

Four themes came to the fore, and the results will focus on the specific theme support measures for the next of kin (Fig 1).

The categories that emerged under the theme of stages after the unexpected loss of a family member, were related to

1. Communicating and speaking out as part of emotional support,

2. Financial assistance,

3. Social worker availability,

4. The realisation of stumbling blocks and new possibilities,

5. Own role in providing hope for the future, and

6. counselling.

### Theme: Support measures for the next of kin

**Communicating and speaking out as part of emotional support.**   The cycle of life involves birth and death in all humans. When a death occurs, the ability to talk about it is associated with fewer grief difficulties and mental health disturbances than in cases where the affected person does not talk about it [20].

However, the stigmatisation of death and dying in some cultures still exist, making it difficult to communicate and having several negative implications. Talking about death helps individuals to work through their fears better [21].

A communicating family has an added element of protection and healing for the psychological health of its members [22].

This supports the assumption that shaping perceptions can be done though focusing away on unpleasant aspects.

One participant was negative in that she had not been part of the process of her mother's death, since her mother had not shared her condition and the reasons for her deterioration with her:

*"Yho . . . Okay. Since I couldn't share things with my mom, I ended up thinking a lot. Thinking negatively. I even ended up not going to school that time, I think, for weeks, and I didn't have a reason. I was not myself. I was hurt. That's all I can say".*

| Themes |
|---|
| Time- related circumstances define behaviour to manage death of a family member |
| Youth go through different stages after the unexpected loss of a family member |
| Managing difficult changes in daily lives of the next of kin |
| Support measures for the next of kin |

**Fig 1. Overall themes of the study.**

*(P5, female, 19 years old)*

Advice from one of the participants was to communicate to a family member or friends or to seek professional help for emotional support:

*"Okay especially from me, I would advise them to speak out. Speak to someone, its either your friends, family member, or if you don't have anyone that you trust then seek for professional assistance. Talk to a social worker, psychologist that they provide at your local clinic. And also–ja, that, mostly".* (Demonstrates 'speaking out' with hands).

*(P1, female, 24 years old)*.

Close relationships with peers also provide support for children, adolescents, and young adults and is effective in helping grieving persons come to terms with their loss [23]. It is important to seek supportive assistance after the loss of a loved one to help prevent prolonged suffering [24].

A participant revealed that self-confidence in speaking out is therapeutic as it helps decrease the burden of keeping things to oneself:

*"I feel like when you speak out, you don't bottle things in; it's like the more you speak about it, the more you feel better about it. Especially if you talk to someone who understands, like a professional assistance. I feel it's good for one when you are in that moment when you recently lost a family member".*

*(P1, female, 24 years old)*

According to Peterson [25], self-confidence is the strength to know oneself, accept oneself, and act on one's convictions. Self-confidence can be seen as a positive feeling about oneself and the world that leads to courageous actions born out of a sense of self-respect [25]. This is a natural rewarding strategy.

In a study conducted by Aoun, Breen, Rumbold, Christian, Same and Abel [26], participants report that professional assistance helped when they experienced grief—being able to talk freely helped. However, some complained of a need for more feedback from professionals.

Participants recognised the value of communication in support groups and the advice of professional counsellors:

*"Hmm . . . I'd say also, the youth. Seek for kind of support that would help in terms of speaking out, being around people with the same experiences would help. For example, support groups where I can sit around and talk with the youth that is also affected".*

*(P1, female, 24 years old)*

*"Like having to talk to someone about what you are going through. And then them giving back advices".*

*(P11, female, 25 years old)*

A study by Aoun et al. [26] reveals that taking the bold step of joining a support group during the grief stage after a loss can give one emotional strength and support as well as needed information on overcoming grief.

Having no one to communicate with can lead to suffering, participants revealed.

*"Then we were suffering with almost everything in the house because we had no one to talk to"*.

(P9, female, 21 years old)

*"I think it's to communicate, but since I've lost my grandmother, I have no one to communicate"*.

(P5, female, 19 years old)

There are numerous families whose usual way of functioning is to talk about only a few issues, feelings, and suppositions. Schwartz [27] says that in such families when any sign of conflict arises, everybody "closes down" or "stuffs it". Schwartz [27] defines 'stuffing it' as keeping one's thoughts and feelings to oneself so as not to affect the emotions of other individuals in the family. In such families, strife is labelled as perilous and harmful [27].

When parents refuse to take time to listen to their children, giving them no safe outlet for their emotions, the child can resort to destructive behaviour.

*"Now I'm living with my mom since my grandmother have died. Since ever I lived with my mom, I have never communicated like serious issues with her because my mom is strict, and she is not that kind of a person who. . .She is not like my grandmother. She cannot talk stuff with us because she thinks we are young. And, at home we are scared sharing things with her. So, we rather keep it in ourselves. That leads to stress. I even end up smoking, I thought it's a better way of living"*. (Pulls fingers).

(P5, female, 19 years old).

It has been shown that girls of unloving and emotionally 'unattuned' mothers share common characteristics. The unmet need for maternal warmth and approval negatively affects their sense of self, making them gravitate to undesirable relationships, and can shape them in ways that are both seen and unseen [28].

Amongst some participants, there was a sense of great faith in God and a strongly felt need to attend church.

*"The parent of my friend invited me to church, one day took me to church. I feel free there and I make new friends and so I joined activities there. I ended up enjoying to go to church even alone. Of my friend is not going to church I go alone because I know I have friends there. I have people to talk to"*.

(P10, female, 25 years old)

A study by McDuffie [29] suggests that social support is precious in African American communities, where it is often coupled with the spiritual or religious component of life.

**Financial assistance.** The government of South Africa has invested time and resources in social development, prioritising basic human needs by providing social grants for financial assistance, healthcare and primary education [30].

A participant spoke of the need for financial assistance, whether through finding a permanent job, from a social worker, or in the form of food support.

*"So, financially assistance would be good, even if I can get a permanent job so that I will be able to take care of my younger siblings. If it's not a job, then financially assistance from the social workers. Food support also, stuff like that".*

(P1, female, 24 years old)

A study conducted by Apelian and Nesteruk [23] reveals that those who have lost a family member who took care of the family's needs are often faced with financial distress after they passing away. As a result, older siblings must seek employment to support their younger siblings.

One participant mentioned receiving support from church members:

*"Some supporting financially where they can. Like buying shoes for the children or buying some sort of groceries. That kind of support".*

(P11, *female, 25 years old*)

Access to food is precarious for many, given the high unemployment rate. In desperation, many individuals and families are looking for jobs in urban areas, but their searches are only sometimes successful, and all members of families become dependent on social grants [30].

A participant who had not yet accessed the foster care grant from the government mentioned how helpful it would be:

*"I think the, I think the foster care . . . at least we manage to go to school, we manage to buy clothes, buy uniform".* (Uses figure tips to count).

(P3, female, 25 years old)

Research studies have shown the critical role and value that the social support grant plays in poor households for improving food security and nutrition and its positive educational effects [31].

In the Republic of South Africa, the Social Assistance Act of 2004 makes provision for seven social support grants; the old age grant, the child support grant, the disability grant, the foster care grant, the war veteran's grant, the care dependency grant, and the grant in aid [30].

The establishment of the child support grant in 1998 was intended to support the children of South Africa. The person responsible for receiving a gift for the child should live in South Africa and earn R4000 or less to qualify [30, 32].

This confirms the assumption that better situations become naturally rewarding.

A participant's current job was low paying, and she struggled to support herself and her siblings. She had to register for a foster care grant from the government.

*"I am old, mos, so we get that money for foster care, they call it foster care when you don't have any parents that is working and you not working, even if you are working but your money is less than R3500 you are able to get that money".* (Shakes).

(P2, female, 25 years old)

A participant believed that the social grant would help her to pursue studies at university:

*"Support, I think also social grant. (Scratches head.) So that I could go study further in varsity. And, even at home I am the older one and we are living with my mom, there are four of us. At*

*home we are depending on her, she is working. It's a one-year contract so after that contract we will be depending on the grant of the two children who are coming after me".*

(P2, female, 25 years old)

The access to a social grant described in research conducted by Bonilla, Zarzur, Handa, Nowlin, Peterman, Ring and Seidenfeld [33], indicated positive effects on the financial empowerment of women, giving them a measure of control over their lives and future.

A participant reported that she and her siblings depended on the grant of their grandmother, as their grant was received by their mother, who kept it for herself:

*"We were depending on her grant. We had a social grant, but our mother was not sending our grants every day, like every month. Sometimes they will not send it and we will depend on our grandmother's social grant".* (Head tilts to the side).

(P5, female, 19 years old)

Studies have shown that the child social grant meant to support children's growth, development, and food requirements are often misused by the recipient or the caregivers of the children, which exacerbates poverty in the home where the children reside [34].

**Social worker availability.** The services of social workers are indispensable for families under tremendous stress, as they facilitate communication and act as a social support structure. A relationship with a caring social worker can tremendously positively affect individuals, helping to ease grief and enabling people to move on. Social workers may engage in one-on-one sessions with children and other family members [35].

A participant mentioned that she would appreciate the support of a social worker.

*Social worker. . . someone who can look after us. Someone who can come and check on us every day that we are okay and give us support so that we won't stress ourselves and think about her. To give us strength to be strong and to have faith that one day they will be someone who will do the same like this. (Looks very worried.)*

*(P4, female, 24 years old)*

Messam and Hart [36] state that strong bonds with a loved one who has passed away may result in a level of grief that requires professional assistance such as counselling by a professional. Although social workers are not trained psychologists, they can fulfil this kind of role in the lives of people left behind, particularly in the case of households headed by very young people.

A participant had to access a social grant provided by the government to support her siblings and son.

*"It was tough unless we go to the social service to get foster care . . . social grant . . . we did get the foster care there because we don't have parents, it's only me, the siblings and my son".* (Palms open, facing upward.)

(P2, female, 25 years old)

South Africa has perhaps the most all-encompassing social protection system on the continent [37] since it involves seven grants and access to social workers.

The ability to recognise one's need for counselling is valuable. Some people may not realise this need and try to push on without help. It seems that young women are aware of their emotional needs and that this is often key to receiving much-needed assistance. A participant mentioned the value of counselling and indicated plans to visit a social worker for support in dealing with stress and accepting the loss of her sister:

*"I think I can go to counselling to support myself because even now, I still can't accept that my sister is no more. Go to visit social workers".* (Eyes fill with tears.)

(P9, female, 21 years old)

The primary responsibility of social workers for clients in grief is to provide individual support and counselling. Grief support provided by the social worker may also include the suggestion of group support and other resources in the community [38].

The findings support the assumption that supports help with behavioural change and enable the individual to manage trauma better.

**The realisation of stumbling blocks and new possibilities.** According to Winter et al., [35] studies in social worker processes have shown that even after a single contact session with a skilled social worker, children and families show changes in their behaviour stemming from changes in their thinking. Counselling enables people to come to a new understanding of themselves and to see possibilities for the future rather than being overwhelmed by grief. This ability to begin to see the future through fresh eyes is particularly pronounced in youth and young adults.

A participant showed insight into the kinds of behaviour and thinking that would enable her to move on from her current unhappiness:

*"Not to over think about it . . . hmm . . . be humble, man, with it, so that I can know how life is, I've been too much disappointed".*

(P2, female, 25 years old)

This participant showed that she was eager to let go of the grief and to continue living with hope and strength:

*"Strength is when someone always giving always power to have that thing that makes you feel "I am okay now," and let me move on, and just accept it in my heart that she is no more and she won't come back. If there is no someone that is going to give me that strength or tell us, "You are going to be okay, and this is not just happening to you only, it's happening on everyone". You are not here in the world to stay forever; we are here to visit. So here in the world we must be strong and move on with life and live just like everyone lives".*

(P2, female, 25 years old)

Factors that promote positive outcomes lie primarily within the individual. These factors include a strong self-concept, bonding with a caregiver and the capacity to think about the experience in a positive manner [39].

A participant seemed to use the positive self-talk spoken of by Clark [40] when she said what she needed to do.

*"Just want to say if you lose someone that you love, it's not the end of the world. You must pull up your socks, go out there and look for a job. So that you can manage to build the others, so that they can see that everything happens for a reason".*

(P2, female, 25 years old)

When individuals permit themselves to be open to new possibilities, they tend to become more positive and can, in the end, accomplish what others might think is incomprehensible. Social relationships and a focus on others help too. Meier [41] states that when people motivate others, they, in turn, become more motivated.

After the passing of her sister, a participant realised the importance of education to get a job:

*"After passing away from my sister, I told myself that I won't go to school anymore. I realise that if I don't go to school, I won't get jobs easily. So, I rather go to study, go further with my study and not focus on the past and all the things, but going up".*

(P9, *female, 21 years old*)

A participant mentioned that at first, living with HIV/AIDS seemed to take away her freedom to live with motivation and inspiration, one can accomplish something her peers did. Still, as time went by, she developed the ability to focus on things that helped her family:

*"I couldn't live the life of the other children . . . people my age. I couldn't do the things that people my age did. Like going out with friends, for instance, it's better now. Like studying, I had to work to focus myself on the kids and what they will be eating".*

(P11, *female, 25 years old*)

This participant was pleased with the support she had received from her church, simply talking to someone you care for:

*"I spoke to people and church members. I also receive support from church members".* (Smiling).

(P11, *female, 25 years old*)

According to Smith and Segal [11], simply talking to somebody else one cares about can assist people in dealing with grief. Some people feel embarrassed at their own need for help, but those who overcome this embarrassment are more likely to find healing than those who bottled up their feelings. Some received valuable social and practical support from her fellow church members.

The findings confirm that moving towards an individual's involvement in activities assists in self-management to well-being.

**Own role in providing hope for the future.**   Jack [42] states that people sometimes become their own 'stumbling block'. It takes work to look inward, see where one's attitudes may be hampering progress, and change one's perspective.

Attending a self-help group can help individuals pursue life changes. Nordmark, Landstad and Hedlund [43] state that self-help is based on people sharing experiences with peers who have experienced similar circumstances. The goal is to achieve new outcomes and regain lost hope for the future through making sound decisions. Self-evaluation, self-development and self-intervention form part of this process. Self-help groups can involve spiritual guidance, the enhancement of self-confidence, encouragement, and reflection on personal values.

The difficulties that participants had been through had the effect of making them stronger. A contributor to this strengthening process was that they had others to take care of–family

members who depended on them. Becoming the breadwinner caused participants to make sacrifices for those who needed them. One said:

*"So that they can know that I will never leave them alone, I will always be there for them. Even no matter how hard, but I'm there. Do you understand?"*

(P2, female, 25 years old)

Sonnenberg [44] states that every relationship requires a certain level of sacrifice to remain healthy. P2 showed great awareness that others looked to her to stay strong, and she seemed to be doing her best to forget the past and move forward in a way that included the needs of others around her.

"I like not to think about my aunt because the more I think about her, the more that. . . (Takes deep breath.) Like I did see that the world is too much for me, I am the only person that is older than them. They are looking up to me. When they see me, they must have hope that everything is going to be alright because we do have the big sister now, I am the only one that is old. The other one is 18, and the other one is 10 and then my son is 7". *(Bites nails).*

(P2, female, 25 years old)

Sonnenberg [44] explains that giving birth to children is not the same as being a parent. A parent can be a guardian or caregiver entrusted with the role of raising children, who, in many cases, must sacrifice a great deal to grow strong, and optimistic young people who have the confidence to face the world and overcome its many obstacles.

The experience of parental death leaves siblings leaning more on one another for support. The relationship among the siblings can become more robust due to the experience of shared pain. Older siblings can take the role of becoming the head and support for the younger ones [23].

At the same time, the burden on the oldest sibling can be significant. A participant spoke of her dreams of academic achievement vanishing because she had to become the head of the family:

*"The only thing I want to say is that it has been a struggle because I couldn't focus on myself. I had to do everything for the children. Even like having . . . I had dreams of going to varsity, but none of that happened because I had to do something that would bring food to the table. So, life isn't about me anymore; it's about them. I wanted to be something in my life. But none of that has happened".* (Voice shakes, about to cry).

(P11, *female, 25 years old*)

The death of a close family member necessitated a massive shift in priorities for most participants. All seemed to be struggling, but many revealed glimpses of the strength that they were seeking. A participant seemed to know the right attitude to take.

*"There's something I need. I as me, I always want to give myself as this person as I am and I want to make myself strong, strong and strong. I just want to encourage other people so that when there is someone pass away with this, so that she could be strong and not blame him or herself about this. Because it's not about someone's fault, it's just an illness that we could accept in oneself that one day we can't fight this illness for long time. Always make sure you*

*eat healthy, exercise every day. Always talk and encourage other people about this illness, go to the clinic and check that you are okay".* (Sad face while sharing.)

(P4, *female, 24 years old*)

It was concluded that focusing away from unpleasant memories, refocusing one on more rewarding aspects.

**Counselling.**   The death of a family member is a painful experience and has the potential to have of long-term impact on the psychological health of the remaining family members. Some turn to self-blame, substance abuse, emotional eating and behavioural disengagement. Death also increases depression, stress, and anxiety [45]. However, when individuals receive grief counselling in the bereavement process, they are greatly helped to adapt to the loss and resolve their grief [45].

The kind of support the youth in this study seemed most in need of, and which they valued most, was emotional support.

*"Oh . . . just counselling. I think counselling is very good. Because at counselling we get to talk about lots of things, how are you feeling, and how does it affect you".*

(P6, *female, 23 years old*)

A participant felt that she needed counselling. She said that the counselling sessions would help her cope and relieve her stress her related to grief after the loss of a family member.

*"I can be supported with . . . uh . . . maybe going to counselling so that I can help relieve my stress".*

(P2, *female, 25 years old*)

A participant spoke of in an informal group of her peers who helped by counselling her:

*"I was hurt. My mother talk to me and take me to other people so that ndizoyeka uku stresser ('I'll stop stressing'). Those people whom you go to when you want to talk about that. There is a group in my stress that when you are stressed, and you want to talk about what you are stressing about. There is a group of people that is doing that project. So, I was going there to talk about what I was stressing about. Then they counselled me. They counselled me, after two to three weeks ndaye nda right* (I became alright)." (Sad face).

(P7, *female, 22 years old*)

Group counselling sessions have a positive role in the stress reduction process [46]. The experience of joining a professional counselling group is reported to be necessary after the loss of a family member, helping a lot with stress relief and practical methods of coping to avoid mental health problems [47].

A participant showed awareness of her own need for guidance. She was concerned about the lack of support she received after her grandmother passed away and expressed a need for someone to help with both the practical aspects of life, such as applying for university, and with the emotional side of her life.

*"Like, I'm a girl. I've got my needs. I am dating. I need someone to be my guide. Like someone to tell me what to do, what not. And someone who will push me to do the right things. Like*

*now I'm supposed to be in varsity, but I had no one to help me with that stuff of going to varsity".* (Uses fingers to count).

(P5, female, 19 years old)

This quote confirms the assumption that obtaining self-efficacy during self-management can improve one's well-being.

Parents who neglect their children for support may be narcissistic. Määttä and Uusiautti [48] explain narcissism, as adapted from Sigmund Freud; a narcissistic parent is cold, rejecting, and indifferent to the needs of others, including their children. They have no emotional reserves to help others and care only about themselves. A narcissistic mother can result in dysfunctional interactions between mother and children [48].

Individuals who attend counselling during stressful times have reported a positive experience that has helped them improve their resilience. A study conducted by Esfandiari, Faramarzi, Amiri, Parsian, Chehrazi, Pasha, Omidvar and Gholinia [49] show that counselling also can increase knowledge and compliance with health-promoting activities.

## Discussion

A death in the family causes many challenges to those left behind, exacerbated if the deceased were a breadwinner. Evidence from the participants showed that participants had to cope not only with the emotional pain of loss and with the practical effects of being left without a source of income. The evidence also shows that many resorted to unhealthy behaviours such as drinking and smoking to reduce stress.

One's belief system plays a role in one's recovery from the loss of a loved one [50]. Sometimes, beliefs help people during suffering or hardship, bringing a sense of control or power. However, Zed [50] points out that sometimes beliefs can be a preventative to personal growth and development. This is assumed to be promoted by constructive thoughts.

Life presents challenges to all. However, with the necessary guidance and support, people can overcome significant obstacles and achieve victory over all sorts of circumstances, including grief, anxiety, and financial hardship. For financial hardship, especially, education is key. Education helps people live with understanding and purpose, opening the mind to conceive new concepts and ways of doing things. Education also allows people to make wiser life choices [51]. Cook [52] points out that with motivation and inspiration, one can accomplish something.

Most young women interviewed expressed shock and denial when they heard their loved one had passed away from a cause they had kept hidden. This created mixed emotions, including anger, remorse and guilt. The loss of a family member leaves relatives struggling to remain stable and realise that there is still a future for them, despite their financial needs and the added stress of conflicts in the family. A few of the participants in the study reported that their lived experience they had attended community self-help sessions, seen a social worker at the local clinic, or spoken to someone at church to deal with the situations they were facing. It is clear from the participants' comments that many were aware of their need for counselling and felt that professional counselling would help them.

### Recommendations

These recommendations aim to ensure the delivery of services such as social work services and counselling to these youths. People cope in different ways. Spiritual coping can lead to a griever's transformation and instil hope during deep pain. It has even been proven that spiritual

coping has helped with chronic illness. A compassionate counsellor can help those who have lost family members cope with their current situation [53].

As part of the Behavioural-focused approach, the youths should share their experience about what they went through. By doing so, they can be enabled to create their youth group to support other child. Research studies have emphasised the importance of establishing safety, trust, and security through being with others who have experienced similar trauma [54].

Youths should join church or community youth groups to talk about death since talking about death help individuals work through their fears. According to Strachan-Proudfit [54], pastors can intentionally incorporate stories of grief into their sermons since many of the stories in the Bible are espoused with real-life losses and levels of hope. They also need encouragement to talk to someone they trust. This can help them to cope with their emotions. Although pastors are not trained grief counsellors, "they can support individuals through listening to their grief and give appropriate advice [54].

Youths should seek supportive assistance provided by healthcare facilities to help them to deal with emotions they face during the death of a family member as part of the *cognitive behavioural approac*h. Youths should attend stress reduction counselling *group sessions*. Grief counselling support groups are a very effective method of handling loss. The youths should seek the help of the healthcare facility's social worker to help them realise new possibilities. They should identify churches in the community that support those who don't have parents and locate feeding schemes. Studies reveal that most people touched by grief are on the fence or stuck, not knowing where to find support [54].

## Recommendations for nursing education and nursing research

Nurses must work on their development and equip themselves with new knowledge by reading articles on the self-management of youths, counselling and social work services and their benefits to those in need. Learning opportunities regarding counselling and providing therapeutic sessions to grieving individuals and families should be provided to nurses. This will increase their knowledge of handling grief and delivering the best care for grieving.

The researcher recommends that further qualitative research be conducted on the social impact on the youth who are affected by death related to HIV/AIDS. A similar study is conducted with male participants. Males were absent at the clinic.

## Limitations of the study

The possible limitation of this study is that all the participants were females, even though the inclusion criteria included both males and females who had lost a family member to HIV/AIDS.

## Conclusion

Support measures for the next of kin were needed. Communicating and speaking out were mentioned as a method of gaining emotional support. Grieving affected the emotions of an individual who experienced the burden of not having someone to speak to about their feelings. Youths did not know how to manage themselves due to a lack of therapeutic sessions, which could provide emotional support and help them with their healing process.

## Supporting information

**S1 File. Ethical clearance.**
(PDF)

**S2 File. Ethical clearance.**
(PDF)

## Acknowledgments

Gratitude is given to youth who partook in the study and Researchpal, who assisted with the open coding of the data.

**Disclaimer:** The views expressed in the submitted article are his or her own.

## Author Contributions

**Conceptualization:** Siphesihle Delani Hlophe, Karien Jooste.

**Data curation:** Siphesihle Delani Hlophe, Karien Jooste.

**Investigation:** Siphesihle Delani Hlophe.

**Methodology:** Siphesihle Delani Hlophe, Karien Jooste.

**Project administration:** Siphesihle Delani Hlophe, Karien Jooste.

**Writing – original draft:** Siphesihle Delani Hlophe, Karien Jooste.

**Writing – review & editing:** Siphesihle Delani Hlophe, Karien Jooste.

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
