## [Decision Letter · Decision Letter 0]

28 Nov 2022

PONE-D-22-11976Support measures for the next of kin who has experienced the unexpected loss of a family member to HIV/AIDSPLOS ONE

Dear Dr. Hlophe,

Thank you for submitting your manuscript to PLOS ONE. After careful consideration, we feel that it has merit but does not fully meet PLOS ONE’s publication criteria as it currently stands. Therefore, we invite you to submit a revised version of the manuscript that addresses the points raised during the review process.

We look forward to receiving your revised manuscript.

Kind regards,

Gianpiero Greco

Academic Editor

PLOS ONE

Journal Requirements:

“no”

“no”

5. Please include your tables as part of your main manuscript and remove the individual files. Please note that supplementary tables (should remain/ be uploaded) as separate "supporting information" files.

Reviewers' comments:

Reviewer's Responses to Questions

**Comments to the Author**

1. Is the manuscript technically sound, and do the data support the conclusions?

Reviewer #1: Yes

2. Has the statistical analysis been performed appropriately and rigorously? 

Reviewer #1: N/A

3. Have the authors made all data underlying the findings in their manuscript fully available?

Reviewer #1: Yes

4. Is the manuscript presented in an intelligible fashion and written in standard English?

Reviewer #1: Yes

5. Review Comments to the Author

Reviewer #1: The range of "youth" age group should be explained and justified in the method section (for example, based on what references?)

Some comparison statement in findings could be re-organized and written into the discussion section.

6. PLOS authors have the option to publish the peer review history of their article (what does this mean?). If published, this will include your full peer review and any attached files.

Reviewer #1: No

---

## [Author Response · Author response to Decision Letter 0]

27 Dec 2022

The manuscript was revised to meet PLOS ONE style templates as per the link provided.

The authors received no specific funding for this work

The authors have declared that no competing interests exist.

Competing Interests has been completed online on the submission form.

The authors received no specific funding for this work. The amended statements have been included in the cover letter.

There are no restrictions to accessing study data; the URL to the repository is as below and is accessible by the public:

https://etd.cput.ac.za/handle/20.500.11838/3370

Research Ethics Committee: Ms Nomathemba Seth, SethN@cput.ac.za Prof Karien Jooste joosteka@cput.ac.za

Tel: +27 21 959 6523

Table included as part of the main manuscript and individual files removed.

Supporting Information files updated on the manuscript accordingly.

Reference have been reviewed to ensure that all links and DOIs are working.

The youth age range has been added and explained int the introduction if the manuscript.

Some comparison statements in the findings were re-organized to the discussion section.

---

## [Editor Report · Decision Letter 1]

20 Mar 2023

Support measures for the next of kin who has experienced the unexpected loss of a family member to HIV/AIDS

PONE-D-22-11976R1

Dear Dr. Hlophe,

We’re pleased to inform you that your manuscript has been judged scientifically suitable for publication and will be formally accepted for publication once it meets all outstanding technical requirements.

Kind regards,

Gianpiero Greco

Academic Editor

PLOS ONE
---

## [Editor Report · Acceptance letter]

30 Mar 2023

PONE-D-22-11976R1 

Support measures for the next of kin who has experienced the unexpected loss of a family member to HIV/AIDS 

Dear Dr. Hlophe:

I'm pleased to inform you that your manuscript has been deemed suitable for publication in PLOS ONE. Congratulations! Your manuscript is now with our production department. 

Kind regards, 

on behalf of

Dr. Gianpiero Greco 

Academic Editor

PLOS ONE